# Cholinesterases Inhibition, Anticancer and Antioxidant Activity of Novel Benzoxazole and Naphthoxazole Analogs

**DOI:** 10.3390/molecules27238511

**Published:** 2022-12-03

**Authors:** Alicja Skrzypek, Monika Karpińska, Małgorzata Juszczak, Aneta Grabarska, Joanna Wietrzyk, Elżbieta Krajewska-Kułak, Marek Studziński, Tadeusz Paszko, Joanna Matysiak

**Affiliations:** 1Department of Chemistry, University of Life Sciences, Akademicka 15, 20-950 Lublin, Poland; 2Institute of Industrial Organic Chemistry, Łukasiewicz Research Network, Annopol 6, 03-236 Warsaw, Poland; 3Department of Medical Biology, Institute of Rural Health, Jaczewskiego 2, 20-090 Lublin, Poland; 4Department of Biochemistry and Molecular Biology, Medical University, Chodźki 1, 20-090 Lublin, Poland; 5Department of Experimental Oncology, Institute of Immunology and Experimental Therapy, Polish Academy of Sciences, R. Weigla 12, 53-114 Wroclaw, Poland; 6Department of Integrated Medical Care, Medical University, M. Skłodowska-Curie 7A, 15-094 Białystok, Poland; 7Department of Physical Chemistry, Institute of Chemistry, Faculty of Chemistry, Maria Curie-Skłodowska University, Pl. M. C. Skłodowkiej 3, 20-031 Lublin, Poland

**Keywords:** benzoxazole, naphthoxazole, acetylcholinesterase, butyrylcholinesterase, antiproliferative activity, antioxidant

## Abstract

Benzoxazole and naphthoxazole fused systems are found in many biologically active molecules. Novel benzoxazole and naphthoxazole analogs functionalized by the 2,4-dihydroxyphenyl moiety were designed, obtained and evaluated as a broad spectrum of biological potency compounds. Sulfinylbis[(2,4-dihydroxyphenyl)methanethione] or its analogs and 2-aminophenols or 1-amino-2-naphthol were used as starting reagents. 4-(Naphtho[1,2-*d*][1,3]oxazol-2-yl)benzene-1,3-diol was identified as the most promising compound of the nanomolar activity against AChE (IC_50_ = 58 nM) of the mixed-type inhibition and of the moderate activity against BChE (IC_50_ = 981 nM). The higher antiproliferative potency against a panel of human cancer cell lines for naphtho[1,2-*d*][1,3]oxazoles than for benzoxazoles was found. The activity of the analog with chlorine atom was in the range of 2.18–2.89 µM (IC_50_) against all studied cells and it is similar to that of cisplatin studied comparatively. Moreover, this compound was not toxic at this concentration to human normal breast cells and keratinocytes. For some compounds it also has proved antioxidant properties at the level of IC_50_ = 0.214 µM, for the most active compound. The lipophilicity of all compounds, expressed as log *p* values, is within the range recommended for potential drugs. The biological activity profile of the considered analogs and their lipophilic level justify the search for agents used in AD or in anticancer therapy in this group of compounds.

## 1. Introduction

Benzoxazole and naphthoxazole are major skeletons of a number of synthetic biologically active heterocyclic compounds, as well as natural products [1]. In recent years, a number of their derivatives, characterized by remarkable biological potency, have been designed and obtained. Benzoxazoles exhibit antioxidant [2], antifungal [3], antibacterial [4], antiviral [5], anti-inflammatory and antiproliferative [6], as well as anticancer activities [6,7].

Derivatives of 2-(2-hydroxyphenyl)-bis(benzoxazole) UK-1 (I) and AJI9561 (II) (Figure 1), being natural products isolated from the strain of *Streptomyces*, are known as anticancer agents [8,9]. UK-1 exhibits a significant cytotoxic activity against HeLa, P338, B16; the other, AJI9561 against Jurkat and P388 cancer cells. Many analogs of UK-1 were designed and obtained as potential anticancer agents [10]. Lower activity was found for simplified 2-(2-hydroxyphenyl)benzoxazole III (Figure 1) and its analogs. They showed cytotoxic properties against the breast cancer cells MCF-7 and the lung cancer cell line A549 at the level of 5–20 µM [8,11]. The other derivatives, 2-amino-aryl-7-aryl-benzoxazoles, exhibit prominent cytotoxic effects towards the human A549 lung cancer cells (EC_50_ of 0.4 µM) [12].

In recent years, many hybrids containing a benzoxazole unit with a cytotoxic effect against various human cancer cell lines have been described, and the attempts have been made to define the mechanisms of their action [13,14]. The cytotoxic properties of benzoxazoles against the cancer cells are the results of the inhibition of human topoisomerase II [15], Aurora B kinase [16], other protein kinases [17], cyclooxygenase-2 (COX-2) [18], or the induction of caspase-3 dependent apoptosis [19].

Some benzoxazoles have been described as acetylcholinesterase (AChE) and butyrylcholinesterase (BChE) inhibitors [19,20,21,22]. Recently, new glycosyl derivatives of benzoxazole were described as AChE inhibitors [23]. The most active compounds had IC_50_ values against AChE and BChE at low µM levels [20]. AChE and BChE are enzymes belonging to the group of serine hydrolases. They exhibit peptidase activity which is extremely important in the pathogenesis of Alzheimer’s disease (AD). As a result of the loss of cholinergic neurons, the concentration of AChE in the brain of AD patients decreases significantly, while the activity of BChE increases sharply. The therapy of AD involves the use of AChE inhibitors which prevent the hydrolysis of acetylcholine (ACh), maintaining the supply of this vital neurotransmitter in brain tissues to improve and stabilize the symptoms of dementia [24,25,26,27].

In addition, the involvement of AChE in non-neuronal functions, such as the regulation of cell proliferation, differentiation, and apoptosis, suggests that it might play an important role in the formation of a cancer and serve as a potential therapeutic target in cancer therapy. A potential example of a mechanism relevant to both cancer and neurodegeneration is the non-enzymatic binding of AChE acting at an allosteric site on the nicotinic alpha-7 receptor. Furthermore, the AChE might play a key role in cancer cell migration through a pathway common to both neurodegeneration and carcinogenesis [28,29].

Naphthoxazoles exhibit antioxidant [30] and antimicrobial as well as anti-HIV activities [31]. They interact with the DNA by the intercalative mode [30] or induce oxygenase-1 expression [31].

2-Aminophenol is a typical substrate used to prepare a variety of 2-substituted benzoxazoles [12,14,22]. The condensation reactions with aldehydes, carboxylic acids or their derivatives (esters or chlorides) are conducted in the catalytic presence of strong acids, heteropolyacids [2], or on microwave radiation [32]. Evindar and Batey elaborated a general method for the formation of benzoxazoles via the copper-catalyzed cyclization of *ortho*-haloanilides [33]. Others have used pyrolysis of o-hydroxybenzanilides [10,11].

Prompted by the described findings, new benzoxazoles and naphthoxazoles with two hydroxyl groups on the 2-aryl ring were designed and prepared. Our studies with sulfinylbis[(2,4-dihydroxyphenyl)methanethione] (STB) [34] showed its high cyclizing potential. Therefore, we attempted to use STB with 2-aminophenols and other similar reagents in the cyclization reaction.

The described compounds were designed and evaluated as dual properties—cholinesterases inhibitors and antiproliferative agents. Cholinesterase-inhibitory activity is found in different classes of anticancer agents and this effect is confirmed by the results of our research [29,35,36].

Variously substituted benzimidazoles and 1,3,4-thiadiazoles with 2,4-dihydroxyphenyl substituent exhibit a significant antiproliferative effect against various human cancer cell lines [36,37]. The most attractive compound is characterized by low toxicity against normal cells and by a prominent neuroprotective effect [38]. Benzimidazoles and 1,3,4-thiadiazoles were also found to be strong AChE inhibitors [39,40]. Some benzimidazoleresorcinol conjugates inhibited in vitro self-induced Aβ (1–42) aggregation, and showed antioxidant properties. The molecular modelling studies exhibited, among others, that the hydroxyl groups and the heterocyclic ring participate in the crucial interactions (hydrogen bonds and π-π stacking) with AChE [41]. Therefore, it seems justified to assess the activity of compounds which, instead of the –NH– group, contain an isosteric oxygen atom. Furthermore, oxidative stress contributes to demyelination and axonal damage in neurodegenerative diseases. Thus, it was justified to create resorcinol conjugates as ChEs inhibitors with the additional antioxidant properties [2].

This paper focuses on the synthesis and biological evaluation of new benzoxazoles and naphthoxazoles with the 2,4-dihydroxyphenyl substituent at C-2 of the heterocyclic ring. The compounds were additionally modified at C-6 of the benzoxazole ring, as well as on the resorcinol moiety. The cholinesterases inhibition potential, cytotoxicity against the human cancer cell lines as well normal cells, and antioxidant properties, were determined.

## 2. Results

### 2.1. Chemistry

The target compounds were prepared using a one-step procedure. 2-Aminophenols and sulfinylbis[(2,4-dihydroxyphenyl)methanethione] (STB) or its analogs modified by –Me, –Et, –Cl or –OH were used in the reaction for the preparation of compounds **1**–**7** (Figure 2) [34]. The reaction was conducted through a 2-hydroxythiobenzanilide which cyclized to the benzoxazole ring. For energy reasons, the –OH group of amine do not split-off in the form of an anion, and the reaction with STB leads to the condensed benzoxazole system (Figure 2). 2-Substituted benzanilides are used commonly in the synthesis of benzoxazoles [10,11]. The course of the reaction confirmed the presence of the molecular ion peaks M^+^ in the EI-MS spectra of the compounds and the disappearance of one hydroxyl group of the substrate in the ^1^H NMR spectra. The protons of the unsubstituted benzoxazole system were registered in the area of 7.90–7.74 ppm. The benzoxazole ring substituted in the C-6 position (**2**–**7**) is disclosed in the form of three groups of typical multiplets: two doublets with a coupling constant *J* = 8.6 Hz and 2.2 Hz, which should be assigned to H-C(4) and H-C(7), respectively and as doublet-doublets of *J* = 8.6 and 2.2 Hz H-C(5). The chemical shift of these bands is quite diverse and depends on the electron properties of the substituents. Please see also provided Appendix A. 

1-Amino-2-naphthol hydrochloride was used in the synthesis of naphtho[1,2-*d*][1,3]oxazoles (**8**–**10**) (Figure 3). The analytical data confirmed the assumed structures formation.

### 2.2. Biological Evaluation

The compounds were evaluated by an in vitro assay as the AChE and BChE inhibitors. The studies were carried out using the modified Ellman’s method [42]. Donepezil was used as the reference drug (Table 1). The studied analogs show a better inhibitory activity against AChE than against BChE, with the exception of compound **6**. The strongest inhibition against AChE was observed for compound **8** (IC_50_ = 0.058 µM), which is the simplest form of the newly-synthesized naphthoxazoles. The results show that replacing the hydrogen attached at position 5 of the resorcinol ring with the ethyl group (**9**) or Cl atom (**10**) causes a significant decrease in activity, IC_50_ = 0.516 and 0.694 µM, respectively. The most active compound **8** is also characterized by the highest selectivity for AChE in relation to BChE. The highest potency against BChE for compound **6** was observed.

To confirm the mechanism of AChE inhibition, the most active compound **8** was tested in kinetic studies using UV-visible spectroscopy, which is a fast, inexpensive and sensitive method. The experiment resulted in the Lineweaver–Burk plot [43]. The data and the graphical analysis are shown in Table 2 and Figure 4.

The examination revealed that V_max_ decreased with increasing K_M_ and increasing concentrations of the inhibitor **8**, which indicated that this compound is a mixed-type inhibitor against AChE [44].

To evaluate the antioxidant potential of compounds as radical scavengers versus the stable free radicals, 2,2-diphenylpicrylhydrazyl (DPPH) was applied. Antioxidant properties were expressed as the IC_50_ value—the concentration of compound required for scavenging 50% of DPPH radicals [45]. The investigated analogs being considered have varying antioxidant potentials in the IC_50_ range of 0.214–0.887 µM. However, the activity is significantly weaker compared to the reference compound quercitine. Compounds **3** and **9** showed the highest scavenging activity.

The compounds were also evaluated for their antiproliferative potencies against human cancer cell lines as the potential anticancer agents. All compounds were tested on the HCV29T line (bladder cancer) as the screening line, followed by those the most active (IC_50_ < 10 µM) on T47D (breast cancer), and A549 (non-small cell lung carcinoma), SW707 (rectal adenocarcinoma). The results are presented in Table 3 and expressed as IC_50_ (μM). In that score, the highest potency was shown by the analogs from the naphtho[1,2-*d*][1,3]oxazole group **8**–**10**, particularly by derivative **10** with a chlorine atom on the benzenediol moiety. Its activity is in the range of 2.18–2.89 µM (IC_50_) against all studied cells, which is similar to that of cisplatin studied comparatively.

The structure–activity relationship concerning antiproliferative activity shows that the parent compound **1** possesses the lowest antiproliferative potency. The presence of the substituents at 6-position of the benzoxazole ring, as well as the lipophilic groups on the benzenediol moiety, enhance the biological effect of the compounds. The additional –OH group in the benzenediol ring affects potency adversely (compound **7**). In the group of analogs with the ester substituent (compounds **3**–**7**), the ethyl derivative is characterized by the highest efficiency and the compound with the third hydroxyl group by the weakest one (**7**). Replacement of the benzoxazole ring with naphthoxazole exerts a positive effect on the antiproliferative potency of the compounds, particularly against the HCV29T cells. The presence of chlorine in the benzene diol ring is the most advantageous for antiproliferative activity.

In view of the observed antiproliferative activity, the influence of the tested compounds on the normal human cells—normal mammary gland epiterial cells of the MCF10A line and the normal keratenocytes HaCaT line—was examined (Figure 5). The cells were treated with compounds **3**, **8** and **10** in the concentration range from 0.1 µM to 25 µM for 72 h, after which the cell viability was tested by analysis of the level of released lactate dehydrogenase (LDH method).

It was observed that compound **3** was not toxic to the normal breast cells and the keratinocytres in the concentrations 0.1–10 µM. Moreover, in MCF10A cells in concentrations 5 and 10 µM, as well as in HaCaT cells in concentrations 2.5, 5 and 10 µM, a statistically significant reduction in the lactate dehydrogenase levels was observed. Compound **8** was toxic to the keratinocytes. A statistically significant increase of LDH was observed in the whole tested range of concentrations. On the other hand, the compound had no influence on mammary cells viability in concentrations 0.1–5 µM. Compound **10** at the tested concentration to 25 µM was not toxic to the MCF10A breast cells and the HaCaT keratinocytes. Moreover, a statistically significant reduction in the level of lactate dehydrogenase in mammary cells in concentrations 0.1–5 µM could indicate its protective effect on these cells.

The lipophilicity parameter was estimated for the tested compounds as it is an important feature that influences the pharmacokinetic properties of bioactive compounds [46,47]. Lipophilicity also determines some biological activities, such as the antiproliferative or fungistatic activity [48]. For lipophilicity determination, reversed-phase column chromatography was used, with the octadecyl C-18 stationary phase and MeOH as the organic modifier in the isocratic conditions. The obtained log k_w_ parameters by extrapolation to pure water as the mobile phase are presented in Table 4 [49].

The log *p* values were also calculated using three different algorithms (Table 4). The results show that the weakest antiproliferative activity against cancer cells is shown by the compounds with the lowest and low lipophilicity. On the other hand, the compounds with high lipophilicity are characterized by the strongest antiproliferative activity. Analyzing the values of log P parameters in relation to pharmacokinetics, the compounds have this parameter in the range recommended for potential drugs favourable for the ADME processes after oral administration. The Clog *p* values of the tested compounds are less than 5, according to the Lipinski recommendations, and the Mlog *p* values are in the range from −2.0 to 4, taking into account the Oprea rules [51].

## 3. Discussion

The obtained results confirm the broad spectrum of biological activity of the compounds, taking into account both the literature reports on the biological activity of benzo(naphtho)xazoles [2,20,22] and the results obtained by our team so far—resorcinol-azole conjugates [41,42,48].

The AChE and BChE inhibitory potential of the considered compounds is higher [22] or similar [2] to that reported for other benzoxazole-based compounds. The investigated compounds (except one) show stronger inhibition against AChE than BChE. This is a general trend observed for the azole-resorcinol conjugates studied by us [39,40,41]. In the group of compounds with a CH_3_O(CO)– substituent in the heterocyclic system, the highest activity is demonstrated by the compounds with the lowest lipophilicity, that is, those with an unsubstituted resorcinol system or with an additional –OH group. It should be noted that compounds **6** and **10** with the lowest activity possess Cl atom in position 5 of the resorcinol ring, which may be the basis for the conclusion that this is a conformationally unfavourable arrangement of atom distribution.

The effectiveness of ChE inhibitors in the prevention of neurodegenerative diseases, including AD, is due to the ACh stimulation of muscarinic M1 receptors, which by activating the protein kinase C, increases the α-secretase activity and thus prevents the formation of senile plaques [35]. In the late stages of the Alzheimer’s disease, BChE plays a more important role in comparison to AChE [52]. The high potency against BChE compared to AChE was found only for compound **6**.

The investigated compounds, similarly to other heterocyclic derivatives with resorcinol ring, exhibit antioxidant properties [39]. Among the most active derivatives are compounds with good activity against AChE. Research indicates that oxidative stress (OS) impacts on the development of neurodegenerative diseases, and that there is a relationship between a β-induced oxidative imbalance and elevated levels of byproducts of lipid peroxidation, protein oxidation and DNA/RNA oxidation [53]. Moreover, this contributes to demyelination and axonal damage in neurodegenerative diseases. Therefore, it is justified to create resorcinol conjugates as ChEs inhibitors with the additional antioxidant properties.

The recent strategy for design and synthesis of compounds as potential drugs has focused on creating combinations with multipurpose properties. Compounds with more than a single molecular target have been found to be more effective as anticancer and antidementia agents [29,35].

The considered compounds, similarly to the previously described resorcinol derivatives of azoles and azines, also show significant antiproliferative activity against cancer cells [37,54]. Osmaniye et al. have described that other benzoxazoles show similar antiproliferative activity against A549 and HT29 cells (MTT assay) [14].

The obtained results confirm the beneficial effect of the presence of a chlorine atom in the benzenediol ring on the antiproliferative potential. The additional –OH group adversely affects potency (compound **7**). This is an effect similar to that observed for other heterocyclic derivatives with a modified benzenediol ring [55]. It also indicates the direction of the influence of changes in lipophilicity on activity. The high activity of naphthoxazoles may be due to their high lipophilicity, as well as the presence of planar polyaromatic system in the molecule, indicating a potential intercalating effect of compounds [30].

For compounds with antiproliferative activity against cancer cells, the level of cytotoxicity for normal cells is important. Analyzing the obtained results of antiproliferative activity and the influence of compounds on the viability of normal cells, it was found that the most attractive in this respect are compounds **3** and **10**. They exhibit the highest antiproliferative activity against human cancer cells, and the IC_50_ values obtained for compound **10** in the cancer cells are lower than for the reference drug—cisplatin. Moreover, the compounds are not toxic for human normal breast cells and keratinocytes. What is more, for both compounds in the culture of breast epithelial cells and keratinocytes, a statistically significant increase in the cell viability was observed, which may indicate the protective effect of these compounds.

Taking into account the overall biological potential of the tested compounds, the derivatives of naphthoxazole seem to be of particular interest. These compounds with dual antitumour and cholinesterase-inhibitory properties can be useful both in the treatment of cancer and AD prevention.

## 4. Materials and Methods

### 4.1. Chemistry

#### 4.1.1. General

Melting points (m.p.) were determined using a BÜCHI B-540 (Flawil, Switzerland) melting point apparatus and were uncorrected. The elemental analysis (C, H, N) was performed on a CHNS/O Elemental Analyzer 2400 Series II (Perkin-Elmer, Waltham, MA, USA). The IR spectra were measured with a Perkin-Elmer FT-IR 1725X spectrophotometer (Perkin-Elmer, Waltham, MA, USA) (in KBr) in the range of 600–4000 cm^−1^. The NMR spectra (^1^H NMR) were recorded in DMSO-*d*_6_ using a Mercury 400 (Varian, Palo Alto, CA, USA) or Bruker DRX 500 (Bruker Daltonics, Inc., Billerica, MA, USA). Chemical shifts (δ, ppm) were described in relation to tetramethylsilane (TMS) and coupling constants (*J*) expressed in Hz. The MS spectra (EI, 70 eV) were recorded using the apparatus AMD-604 (AMD Analysis & Technology AG, Harpstedt, Germany).

#### 4.1.2. Synthesis of the Compounds

Procedure for the synthesis of compounds **1**–**7**. A mixture of the corresponding commercially available 2-aminophenol hydrochloride (10 mmol) and STB (10 mmol) in MeOH (80 mL) was heated to reflux for 2.5–3.5 h (Figure 2). The hot mixture was filtered, and subsequently the filtrate was concentrated (**1**, **4**). In the case of compounds **2**, **3** and **5**–**7**, MeOH/H_2_O solution was added to the concentrated filtrate to precipitate the product into a solid state. Finally, the compound was recrystallized from MeOH (compounds **2**, **3**, **5**) (50 mL), from MeOH/H_2_O (1:1) (**1**), or from MeOH/H_2_O (3:1) (**4**, **6**, **7**) (50 mL). STB was obtained according to the previously described procedure [34].

*4-(1,3-Benzoxazol-2-yl)benzene-1,3-diol* (**1**). Yield: 71%; m.p.: 189–190 °C; ^1^H NMR (100 MHz, DMSO-*d*_6_) δ: 11.29 (s, 1 H, C_3_-OH (exchangeable in D_2_O)), 10.42 (s, 1 H, C_1_-OH (exchangeable in D_2_O), 7.90 (m, 1 H, C_5_-H), 7.74 (m, 2 H, C_5′_,_6′_-H), 7.45 (m, 2 H, C_4′_,_7′_-H), 6.58 (m, 1 H, C_6_-H), 6.50 (m, 1 H, C_2_-H) ppm; ^13^C NMR (125 MHz, DMSO-*d*_6_) 162.9, 162.8, 150.8, 148.1, 139.5, 128.8, 125.1, 125.0, 118.5, 110.7, 108.7, 102.8, 101.9 ppm; δ: IR (KBr, cm^−1^): 3367 (OH), 1636 (C=N), 1600 (C=C), 1560 (C=C), 1500 (C=C), 1474, 1456, 1433, 1324, 1296, 1278, 1245, 1229 (C-OH), 1170, 1134, 1108, 1054, 1003, 977, 940, 895, 847, 808, 763, 745, 721; EI-MS (*m*/*z*, %): 227 (M^+^, 100), 192 (3), 170 (21), 163 (5), 158 (3), 130 (9), 99 (3), 63 (3), 52 (2), 39 (2). Anal. Calc. for C_13_H_9_NO_3_ (227.22) (%): C, 68.72; H, 3.99; N, 6.16. Found (%): C, 68.69; H, 4.01; N, 6.14.

*4-(6-Hydroxy-1,3-benzoxazol-2-yl)-2-methylbenzene-1,3-diol* (**2**). Yield: 73%; m.p.: 236–238 °C (R). ^1^H NMR (400 MHz, DMSO-*d*_6_) δ: 11.30 (s, 1H, C_3_-OH), 10.35 (s, 1H, C_1_-OH), 9.87 (s, 1H, C_6′_-OH), 7.78 (d, *J* = 8.6 Hz, 1H, C_5_-H), 7.54 (d, *J* = 8.6 Hz, 1H, C_Ar_-H), 7.11 (d, *J* = 2.1 Hz, 1H, C_Ar_-H), 6.86 (dd, *J* = 8.6 and 2.2 Hz, 1H, C_Ar_-H), 6.52 (dd *J* = 8.6 Hz, 1H, C_6_-H), 2.10 (s, 3H, CH_3_) ppm. IR (KBr, cm^−1^): 3489 (OH), 3260 (OH), 2957 (CH), 2836 (CH), 1643 (C=N), 1618 (C=C), 1562 (C=C), 1503 (C=C), 1489, 1437, 1342, 1319, 1299, 1279, 1260, 1242 (C-OH), 1183, 1135, 1128, 1107, 1051, 977, 957, 843, 807, 760, 718. EI-MS (*m*/*z*, %): 257 (M^+^, 10), 243 (100), 214 (11), 201 (9), 173 (17), 159 (2), 137 (4), 121 (4), 108 (3), 79 (6), 69 (4), 63 (35), 51 (6), 39 (5). Anal. Calc. for C_14_H_11_NO_4_ (257.24) (%): C, 65.37; H, 4.31; N, 5.44. Found (%): C, 64.13; H, 4.29; N, 5.41.

*Methyl 2-(2,4-dihydroxyphenyl)-1,3-benzoxazole-6-carboxylate* (**3**). Yield: 81%; m.p.: 252–254 °C. ^1^H NMR (400 MHz, DMSO-*d*_6_) δ: 11.08 (s, 1H, C_3_-OH), 10.50 (s, 1H, C_1_-OH), 8.26 (d, *J* = 1.0 Hz, 1H, C_7′_-H), 8.02 (dd, *J* = 8.5 and 1.5 Hz, 1H, C_5′_-H), 7.87 (d, *J* = 8.7 Hz, 1H, C_6_-H), 7.85 (d, *J* = 8.3 Hz, 1H, C_4′_-H), 6.55 (dd, *J* = 8.7 and 2.3 Hz, 1H, C_6_-H), 6.48 (d, *J* = 2.2 Hz, 1H, C_2_-H), 3.90 (s, 3H, Me) ppm. ^13^C NMR (125 MHz, DMSO-*d*_6_) 192.1 (C=O), 165.7, 162.0, 156.1, 148.8, 135.9, 132.2, 126.5, 122.7, 119.7, 117.1, 115.2, 108.2, 102.4, 52.0 (CH_3_) ppm; IR (KBr, cm^−1^): 3308 (OH), 2962 (CH), 1697 (C=O), 1636 (C=N), 1621 (C=C), 1598 (C=C), 1559 (C=C), 1433, 1358, 1327, 1302, 1263, 1237 (C-OH), 1198, 1169, 1138, 1122, 1079, 1056, 978, 959, 924, 882, 853, 809, 771, 742, 721. EI-MS (*m*/*z*, %): 285 (M^+^, 100), 254 (69), 199 (3), 127 (7), 65 (2), 63 (4), 39 (2). Anal. Calc. for C_15_H_11_NO_5_ (285.25) (%): C, 63.16; H, 3.89; N, 4.91. Found (%): C, 63.22; H, 3.91; N, 4.88.

*Methyl 2-(2,4-dihydroxy-3-methylphenyl)-1,3-benzoxazole-6-carboxylate* (**4**). Yield: 87%; m.p.: 249–252 °C. ^1^H NMR (400 MHz, DMSO-*d*_6_) δ: 11.40 (s, 1H, C_3_-OH), 10.44 (s, 1H, C_1_-OH), 8.26 (d, *J* = 1.2 Hz, 1H, C_Ar_-H), 8.02 (dd, *J* = 8.3 and 1.5 Hz, 1H, C_Ar_-H), 7.85 (d, *J* = 8.4 Hz, 1H, C_Ar_-H), 7.74 (d, *J* = 8.7 Hz, 1H, C_Ar_-H), 6.64 (d, *J* = 8.7 Hz, 1H, C_Ar_-H), 3.90 (s, 3H, OMe), 2.08 (s, 3H, Me) ppm. IR (KBr, cm^−1^): 3380 (OH), 2963 (CH), 2841 (CH), 1699 (C=O), 1638 (C=N), 1616 (C=N), 1555 (C=C), 1485, 1433, 1360, 1300, 1238 (C-OH), 1203, 1138, 1082, 977, 933, 877, 829, 804, 770, 736. EI-MS (*m*/*z*, %): 299 (M^+^, 100), 285 (2), 70 (13), 268 (22), 254 (5), 242 (2), 12 (7), 150 (2), 120 (2), 63 (2). Anal. Calc. for C_16_H_13_NO_5_ (299.28) (%): C, 64.21; H, 4.38; N, 4.68. Found (%): C, 64.08; H, 4.40; N, 4.69.

*Methyl 2-(5-ethyl-2,4-dihydroxyphenyl)-1,3-benzoxazole-6-carboxylate* (**5**). Yield: 77%, m.p.: 203–206 °C. ^1^H NMR (400 MHz, DMSO-*d*_6_) δ: 10.92 (s, 1H, C_3_-OH), 10.48 (s, 1H, C_1_-OH), 8.25 (d, *J* = 1.0 Hz, 1H, C_7′_-H), 8.01 (dd, *J* = 8.3 and 1.5 Hz, 1H, C_5′_-H), 7.83 (d, *J* = 8.3 Hz, 1H, C_4′_-H), 7.71 (s, 1H, C_5_-H), 6.53 (s, 1H, C_2_-H), 3.90 (s, 3H, OMe), 2.56 (q, *J* = 7.4 Hz, 2H, CH_2_Me), 1.17 (t, *J* = 7.5 Hz, 3H, CH_2_Me) ppm. IR (KBr, cm^−1^): 3382 (OH), 2960 (CH), 2837 (CH), 1698 (C=O), 1638 (C=N), 1621 (C=C), 1548 (C=C), 1511 (C=C), 1436, 1400, 1358, 1302, 1265 (C-OH), 1231, 1198, 1180, 1152, 1080, 1039, 972, 883, 848, 809, 772, 743, 708. EI-MS (*m*/*z*, %): 313 (M^+^, 54), 298 (M^+^-Me, 100), 282 (4), 270 (7), 228 (3), 157 (2), 134 (7), 77 (2), 69 (3), 63 (4). Anal. Calc. for C_17_H_15_NO_5_ (313.30) (%): C, 65.17; H, 4.83; N, 4.47. Found (%): C, 65.23; H, 4.81; N, 4.50.

*Methyl 2-(5-chloro-2,4-dihydroxyphenyl)-1,3-benzoxazole-6-carboxylate* (**6**). Yield: 89%; m.p.: 201–202 °C. ^1^H NMR (400 MHz, DMSO-*d*_6_) δ: 11.39 (s, 1H, C_3_-OH), 11.04 (s, 1H, C_1_-OH), 8.26 (d, 1 Hz, 1H, C_Ar_-H), 8.03 (dd, *J* = 8.3 and 1.5 Hz, 1H, C_Ar_-H), 7.92 (s, 1H, C_Ar_-H), 7.86 (d, *J* = 8.3 Hz, 1H, C_Ar_-H), 6.76 (s, 1H, C_5_-H), 3.91 (s, 3H, OMe) ppm. ^13^C NMR (125 MHz, DMSO-*d*_6_) 190.5 (C=O), 165.5, 158.4, 157.0, 154.2, 148.8, 135.0, 132.0, 126.4, 122.5, 119.7, 115.3, 111.8, 103.7, 52.0 (CH_3_) ppm; IR (KBr, cm^−1^): 3535 (OH), 3478 (OH), 3080 (CH), 2983 (CH), 1708 (C=O), 1624 (C=N), 1605 (C=N), 1583 (C=C), 1551 (C=C), 1508 (C=C), 1474, 1438, 1407, 1352, 1295, 1259, 1230 (C-OH), 1200, 1135, 1080, 1031, 978, 920, 893, 845, 837, 769, 740, 720. EI-MS (*m*/*z*, %): 319 (M^+^, 100), 288 (46), 254 (3), 232 (12), 144 (6), 119 (2), 99 (2), 79 (2), 69 (2), 62 (3), 51 (2). Anal. Calc. for C_15_H_10_ClNO_5_ (319.70) (%): C, 56.35; H, 3.15; N, 4.38. Found (%): C, 56.48; H, 3.17; N, 4.40.

*Methyl 2-(2,3,4-trihydroxyphenyl)-1,3-benzoxazole-6-carboxylate* (**7**). Yield: 67%; m.p.: 203–204 °C. ^1^H NMR (400 MHz, DMSO-*d*_6_) δ: 10.97 (s, 1H, C_3_-OH), 10.03 (s, 1H, C_1_-OH), 8.27 (d, *J* = 1 Hz, 1H, C_Ar_-H), 8.04 (dd, *J* = 8,3 and 1.5 Hz, 1H, C_Ar_-H), 7.87 (d, *J* = 8.3 Hz, 1H, C_Ar_-H), 7.42 (d, *J* = 8.7 Hz, 1H, C_Ar_-H), 6.59 (d, *J* = 8.7 Hz, 1H, C_Ar_-H), 3.90 (s, 3H, OMe) ppm. IR (KBr, cm^−1^): 3425 (OH), 2959 (CH), 2847 (CH), 1703 (C=O), 1638 (C=N), 1622 (C=N), 1563 (C=C), 1500 (C=C), 1483, 1437, 1351, 1298, 1273, 1236 (C-OH), 1139, 1121, 1080, 1025, 982, 957, 923, 900, 879, 840, 805, 770, 742. EI-MS (*m*/*z*, %): 301 (M^+^, 100), 70 (28), 242 (7), 14 (13), 167 (2), 151 (2), 135 (15), 11 (3), 107 (2), 98 (3), 79 (2), 63 (7), 51 (2), 39 (2). Anal. Calc. for C_15_H_11_NO_6_ (301.25) (%): C, 59.80; H, 3.68; N, 4.65. Found (%): C, 56.63; H, 3.70; N, 4.63.

Procedure for the synthesis of compounds **8**–**10**. A mixture of 1-amino-2-naphthol hydrochloride (1.3 mmol) and the corresponding STB (1.3 mmol) in MeOH (7 mL) was heated to reflux for 3.5–4 h (Figure 3). The hot mixture was filtered and the filtrate concentrated. Finally, the compound was recrystallized from MeOH/H_2_O (3:1) (compounds **8**, **9**) or from MeOH/H_2_O (5:1) (5 mL) (**10**).

*4-(Naphtho[1,2-d]*[1,3]*oxazol-2-yl)benzene-1,3-diol* (**8**). Yield: 74%; m.p.: 166–168 °C. ^1^H NMR (400 MHz, DMSO-*d*_6_) δ: 11.45 (s, 1H, C_3_-OH), 10.67 (s, 1H, C_1_-OH), 8.85 (m, 1H, C_Ar_-H), 8.41 (s, 1H, C_Ar_-H), 8.08 (d, *J* = 7.8 Hz, 1H, C_Ar_-H), 7.91 (d, *J* = 7.4 Hz, 2H, C_Ar_-H), 7.87 (m, 1H, C_Ar_-H), 7.75 (m, *J* = 8.2 and 1.4 Hz, 1H, C_Ar_-H), 7.67 (m, *J* = 8.4 and 1.0 Hz, 1H, C_Ar_-H), 6.78 (s, 1H, C_Ar_-H), 6.53 (s, 1H, C_6_-H) ppm. IR (KBr, cm^−1^): 3480 (OH), 3279 (OH), 2923 (CH), 2854 (CH), 1632 (C=N), 1608 (C=N), 1562 (C=C), 1496, 1448, 1437, 1402, 1381, 1333, 1277, 1252 (C-OH), 1234, 1179, 1134, 1082, 1058, 1026, 1005, 975, 940, 882, 843, 799, 776, 719, 702. EI-MS (*m*/*z*, %): 277 (M^+^, 100), 248 (5), 220 (7), 138 (2), 114 (2), 88 (1), 63 (1). Anal. Calc. for C_17_H_12_NO_3_ (277.27) (%): C, 73.64; H, 4.00; N, 5.05. Found (%): C, 73.71; H, 3.99; N, 5.08.

*4-Ethyl-6-(naphtho[1,2-d]*[1,3]*oxazol-2-yl)benzene-1,3-diol* (**9**). Yield: 71%; m.p.: 189–192 °C. ^1^H NMR (400 MHz, DMSO-*d*_6_) δ: 11.10 (s, 1H, C_3_-OH), 10.31 (s, 1H, C_1_-OH), 8.61 (m, 1H, C_Ar_-H), 8,50 (m, 1H, C_Ar_-H), 8,11 (m, 1H, C_Ar_-H), 7.98 (m, 2H, C_Ar_-H), 7.61 (s, 1H, C_Ar_-H), 6.56 (s, 1H, C_6_-H), 2.58 (q, 2H, CH_2_Me), 1,23 (t, 3H, CH_2_Me) ppm. ^13^C NMR (125 Hz): 162.4, 160.3, 157.4, 145.7, 142.8, 134.8, 130.9, 128.7, 127.1, 126.8, 125.5, 124.6, 123.2, 121.6, 110.9, 102.6, 101.7, 22.0, 14.1 ppm. IR (KBr, cm^−1^): 3434 (OH), 2962 (CH), 2924 (CH), 2847 (CH), 1624 (C=N), 1543 (C=C), 1507 (C=C), 1457, 1406, 1383, 1358, 1278, 1242 (C-OH), 1143, 1087, 1061, 1030, 1004, 884, 843, 802, 747. EI-MS (*m*/*z*, %): 305 (M^+^, 62), 290 (M^+^-Me, 100), 276 (4), 262 (13), 248 (3), 220 (11), 204 (6), 178 (13), 145 (6), 114 (8), 110 (7), 91 (4), 80 (14), 44 (14), 40 (37), 36 (5). Anal. Calc. for C_19_H_15_NO_3_ (305.33) (%): C, 74.74; H, 4.96; N, 4.59. Found (%): C, 74.60; H, 4.98; N, 4.57.

*4-Chloro-6-(naphtho[1,2-d]*[1,3]*oxazol-2-yl)benzene-1,3-diol* (**10**). Yield: 73%; m.p.: 228–231 °C. ^1^H NMR (400 MHz, DMSO-*d*_6_) δ: 11.22 (s, 1H, C_3_-OH), 8.85 (m, 1H, C_Ar_-H), 8.45 (m, 1H, C_Ar_-H), 8.13 (d, *J* = 8.1 Hz, 1H, C_Ar_-H), 7.97 (d, *J* = 7.1 Hz, 1H, C_Ar_-H), 7.73 (m, 1H, C_Ar_-H), 7.62 (m, 2H, C_Ar_-H), 6.78 (s, 1H, C_2_-H) ppm. ^13^C NMR (125 MHz, DMSO-*d*_6_) 160.9, 157.5, 157.4, 145.9, 143.6, 134.7, 130.8, 128.7, 127.7, 127.2, 126.4, 126.0, 125.6, 124.6, 121.5, 110.9, 104.2 ppm; IR (KBr, cm^−1^): 3435 (OH), 3107 (OH), 3078 (OH), 2968 (CH), 1633 (C=N), 1601 (C=C), 1506 (C=C), 1483, 1400, 1384, 1342, 1300, 1257 (C-OH), 1222, 1183, 1164, 1091, 1062, 1030, 1005, 938, 883, 842, 792, 742. EI-MS (*m*/*z*, %): 311 (M^+^, 100), 282 (6), 248 (7), 220 (8), 191 (4), 156 (4), 114 (11), 110 (3), 88 (4), 63 (3), 51 (2). Anal. Calc. for C_17_H_10_ClNO_3_ (311.72) (%): C, 65.50; H, 3.23; N, 4.49. Found (%): C, 65.42; H, 3.25; N, 4.51.

#### 4.1.3. Chromatography

HPLC measurements were performed using a liquid chromatograph (Knauer, Berlin, Germany) with a dual pump and a UV-visible detector. The measurements were made at 254 or 320 nm at room temperature. A Eurosil Bioselect C18 (5 μm, 300 × 4.6 mm) column was applied as the stationary phase. The mobile phase consisted of different volume mixtures of 20 mM acetate buffer as the aqueous phase, and MeOH as the organic modifier (pH 7.4). The different concentrations of MeOH from 0.5 to 0.90 (*v*/*v*) range was used, depending on the structure of the compound at 0.05 intervals. The flow rate was 1 mL/min. Retention time of an unretained solute (t_0_) was determined by the injection of a small amount of acetone dissolved in water.

Regular changes in the retention coefficients of compounds in a function of the organic modifier MeOH content in the mobile phase are described by the Soczewiński–Wachtmeister equation [56]:log k = log k_w_ + S φ(1)

φ—the volume fraction of organic modifier (MeOH) in the mobile phase; log k_w_—the intercept; S—slope of the regression curve. Log k_w_ refers to the retention parameter of a compound with pure water as the mobile phase. The S and log k_w_ parameters for the considered compounds determined from the linear dependence (1) are summarized in Table 4.

#### 4.1.4. Computational Methods

The Clog P and Mlog values were calculated using the ChemDraw Ultra 10.0 [57] or MedChem Designer (TM) 3.0.0.30 [58], respectively.

### 4.2. Biological Assays

#### 4.2.1. SRB Cytotoxicity Assay

In the antiproliferative assay in vitro, the following human cell lines were applied: T47D (breast cancer), SW707 (rectal adenocarcinoma), A549 (non-small cell lung carcinoma) from the American Type Culture Collection (Rockville, MD, USA) and HCV29T (bladder cancer) from the Fibiger Institute, Copenhagen, Denmark. Cytotoxicity assay was performed after 72 h exposure of the cultured cells at the concentration ranging from 0.1 to 100 μg/mL of the tested agents. The SRB test measuring the cell proliferation inhibition in the in vitro culture was applied [59]. The cellular material fixed with TCA was stained with 0.4% sulforhodamine B (Chemie GmbH, Steinheim, Germany) dissolved in 1% acetic acid (POCh, Gliwice, Poland) for 30 min. The unbound dye was removed by rinsing (four times) with 1% acetic acid, and the protein-bound dye extracted with 10 mM unbuffered Tris base (tris(hydroxymethyl) aminomethane (POCh, Gliwice, Poland), for determination of optical density (at 540 nm) in a computer-interfaced, 96-well Uniskan II microtiter plate reader (Labsystems, Helsinki, Finland). The experiments were repeated at least 3 times. The IC_50_ values were calculated by Cheburator 0.9.0 software [60]. Details were described in a previous paper [37].

#### 4.2.2. Cytotoxicity Assessment

##### Cell Culture and Treatments

Normal human epithelial cells from the mammary gland, MCF10A (a generous gift from our colleague Dr. Saraswati Sukumar (JHU), who obtained them recently from ATCC), and normal human keratinocytes HaCaT (also obtained from ATCC), were cultured in DMEM/F12 medium (Gibco), supplemented with 5% horse serum (Gibco), 5% FBS (Sigma-Aldrich Chemie GmbH, Taufkirchen, Germany ), penicillin streptomycin solution (1×) (Gibco) at 37 °C and 5% CO_2_. The cells were subcultured twice a week.

##### LDH Assay

Human epiterial breast cells MCF10A and human keratinocytes HaCaT (1 × 10^4^/mL) cells were placed on 96-well plates (Nunc, Roskilde, Denmark). The next day, the cells were exposed to increasing concentrations of compounds **3**, **8** and **10** in 0.1, 0.25, 0.5, 1, 2.5, 5, 10 and 25 μM in fresh culture medium. Cytotoxicity was estimated based on the measurements of cytoplasmic lactate dehydrogenase (LDH) activity released from the damaged cells after the 72 h exposure to the compounds. The LDH assay was performed according to the manufacturer’s instructions (Cytotoxicity Detection KitPLUS LDH, Roche, Mannheim, Germany). Absorbance was measured using a microplate at two different wavelengths: “measurement wavelength” (492 nm) and “reference wavelength” (690 nm), using a microplate [61].

#### 4.2.3. In Vitro AChE and Bche Inhibition Assay

Acetylcholinesterase (AChE, E.C. 3.1.1.7, from human erythrocytes), butyrylcholinesterase (BChE, E.C. 3.1.1.8, from equine serum), acetylthiocholine iodide (ATCh), butylthiocholine iodide (BTCh), 5,5′-dithiobis-(2-nitrobenzoic acid) (DTNB) and donepezil were purchased from Sigma-Aldrich (Steinheim, Germany). The inhibitory activities against AChE and BChE of the newly-synthesized compounds were determined exactly, according to a previous paper [42]. Donepezil was applied as the standard. The compounds were tested in the range from 10^−3^ to 10^−9^ M at 412 nm using a Varian Cary 50 Spectrophotometer. The samples were analysed in triplicate. IC_50_ values were designated from a dose–response curve obtained by plotting the percentage of inhibition versus the log concentration, using GraphPad Prism 8 software. Results were presented as the mean ± standard error of measurements (SEM). AChE and BChE activity data were compared using the Kruskal–Wallis test at the significance level of 1%.

#### 4.2.4. Kinetic Study of AChE Inhibition Assay

Ellman’s method was used to prepare a kinetic characterization of AChE inhibition for compound **8**. Plots of 1/velocity versus the reciprocal substrate concentration were constructed at different final concentrations of the ATCh iodide in the range of 2.5–15 nM. The studies were carried out using the final concentrations of inhibitor **8** (25, 50, 70 nM) and without it. The tested medium contained DTNB solution (0.4 mg/mL), AChE solution (2 U/mL), solution of the examined compound **8**, and different concentrations of the ATCh iodide substrate solution (S). Absorbance was detected at 412 nm at room temperature after 10 min. The obtained data were calculated and presented on the Lineweaver–Burk plot. The determined the Michaelis–Menten constant (K_M_) and the maximum velocity of the enzyme (V_max_) values were used to define the type of enzyme inhibition [39].

#### 4.2.5. DPPH Free Radical Scavenging Activity

The DPPH free radical scavenging activities of the synthesized compounds were measured by DPPH, according to the method described in the paper [41]. The free radical scavenging activity was measured as a decrease in the absorbance of DPPH, and calculated using the equation:DPPH scavenging effects (%) = 100 − [(A_0_ − A_1_)/A_0_]∙100](2)
where: A_0_ was the absorbance of the control reaction and A_1_ absorbance in the presence of the sample of the analysed compounds [62]. The IC_50_ values were calculated as the concentration of compound required for scavenging 50% of DPPH [45]. Results were presented as the mean ± standard deviation (SD). The antioxidant activity data were compared using the one-way ANOVA test at the significance level 5%.

## 5. Conclusions

In conclusion, we were able to devise a new one-step synthesis method of 2-aryl substituted oxazole fused system using 1,2-aminohydroxyaryles and the electrophilic reagent STB. Naphthoxazoles seem to be particularly interesting as the compounds of bi-directional biological activity. The most interesting derivative shows high antiproliferative potency against human cancer cell line and simultaneously is not toxic to normal human cells. Additionally, a protective effect of this compound was confirmed.

The investigated compounds are also cholinesterase inhibitors and exhibit antioxidant properties. Since a lot of evidence points to the active involvement of oxidative stress in neurodegenerative diseases, the compounds with polyphenol rings have also become a source of new therapeutic agents against AD.

Considering the general biological potential of the tested compounds and their lipophilic nature, these compounds with dual antitumour and cholinesterase inhibiting properties may be useful in the design of effective drugs against cancer and AD.

## Figures and Tables

**Figure 1 molecules-27-08511-f001:**
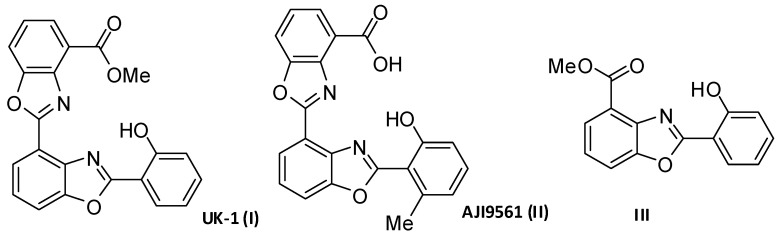
Structures of 2-(2-hydroxyphenyl)benzoxazole analogs as cytotoxic compounds against the human cancer cell lines [10].

**Figure 2 molecules-27-08511-f002:**
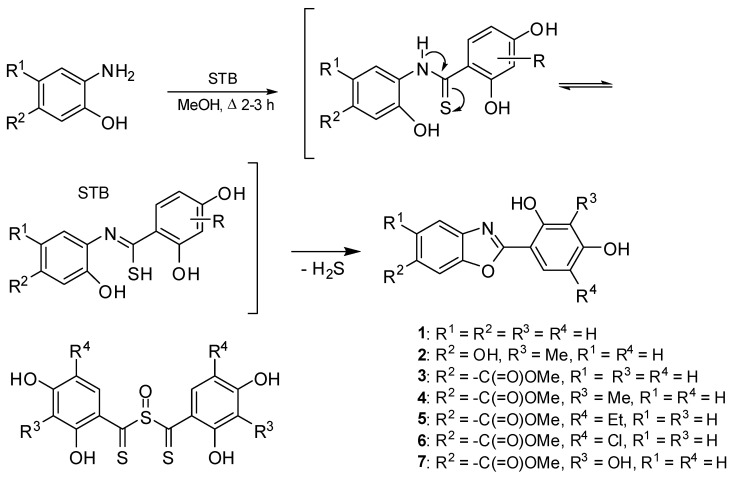
Synthesis scheme of 1,3-benzoxazoles (**1**–**7**).

**Figure 3 molecules-27-08511-f003:**
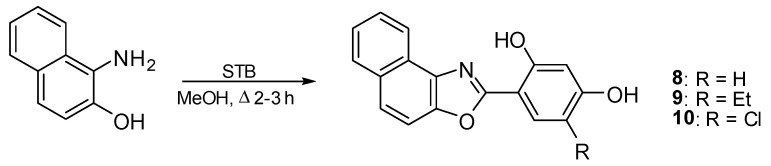
Synthesis scheme of naphtho[1,2-*d*][1,3]oxazoles (**8**–**10**).

**Figure 4 molecules-27-08511-f004:**
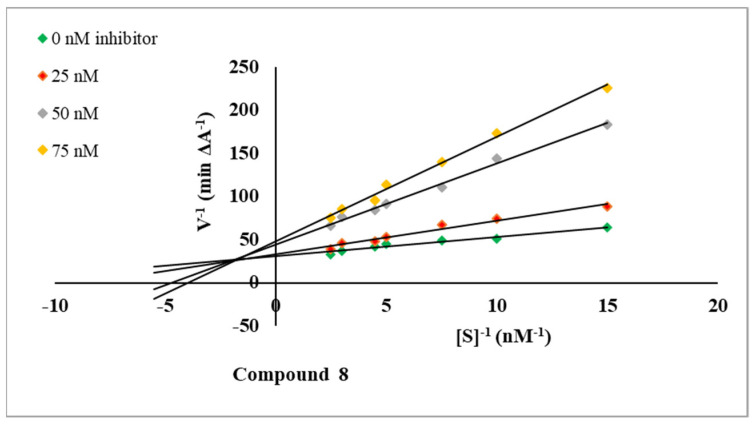
Lineweaver-Burk reciprocal plots for the AChE inhibition by compound **8**.

**Figure 5 molecules-27-08511-f005:**
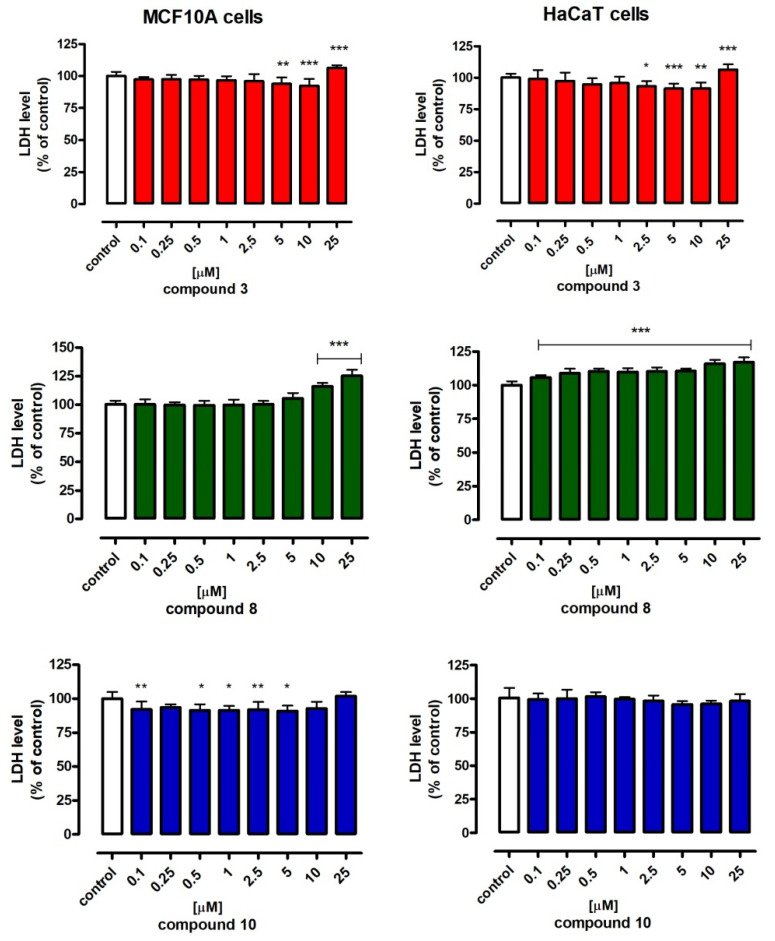
Influence of compounds **3**, **8** and **10** on viability of normal mammary gland MCF10A cells and normal keracinocytes HaCaT cells. Levels of lactic dehydrogenase (LDH) released from cells in the absence (control) and in the presence of tested compounds (0.1–25 μM) were quantified after 72-h treatment. Data represents mean normalized optical densities ± SEM of 4–6 trials, *** at least *p* < 0.001, ** *p* < 0.01, * *p* < 0.05 vs. control, one-way ANOVA test, post hoc Tukey.

**Table 1 molecules-27-08511-t001:** In vitro inhibitory concentration of the new compounds and against AChE and BChE (IC_50_, μM), selectivity, and the radical scavenging effect IC_50_ (DPPH).

Compound	AChE ^1^ IC_50_ [μM]	BChE ^2^ IC_50_ [μM]	Selectivity for AChE ^3^	IC_50_ (DPPH) ^4^ [µM]
**3**	0.113 ± 0.011 *	1.132 ± 0.016 *	10.017	0.286 ± 0.122 **
**6**	0.831 ± 0.014	0.328 ± 0.027	0.394	0.861 ± 0.071
**7**	0.183 ± 0.031	0.667 ± 0.004	3.644	0.887 ± 0.064
**8**	0.058 ± 0.010	0.981 ± 0.015	18.166	0.452 ± 0.031
**9**	0.516 ± 0.009	1.017 ± 0.032	1.971	0.214 ± 0.024
**10**	0.694 ± 0.011	0.887 ± 0.041	1.278	0.633 ± 0.146
donepezil	0.017 ± 0.002	1.159 ± 0.016	68.176	-
quercetin	-	-	-	0.032 ± 0.001

^1^ IC_50_—50% inhibitory concentration (means ± SEM of three independent experiments) of AChE (* *p* < 0.01; Kruskal–Wallis test). ^2^ IC_50_—50% inhibitory concentration (means ± SEM of three independent experiments) of BChE (* *p* < 0.01; Kruskal–Wallis test). ^3^—selectivity for AChE = IC_50_ (BChE)/IC_50_ (AChE). ^4^ IC_50_—50% inhibitory concentration (means ± SD of three experiments) (** *p* < 0.05; one-way ANOVA test).

**Table 2 molecules-27-08511-t002:** The K_M_ and V_max_ values at different compound **8** concentrations in relation to AChE.

Concentration [nM]	K_M_ [nM]	V_max_ [A/min]
75	0.254	0.021
50	0.212	0.023
25	0.117	0.031
0	0.072	0.033

**Table 3 molecules-27-08511-t003:** Antiproliferative activity of compounds **1**–**10** against human cancer cell lines.

Compound	Cell Line/IC_50_ ^1^/[µM]			
	HCV 29T	A549	T47D	SW 707
**1**	33.93 ± 4.77	- ^2^	- ^2^	- ^2^
**2**	8.34 ± 1.53	6.81 ± 2.23	5.28 ± 0.16	4.38 ± 1.57
**3**	6.43 ± 1.12	4.28 ± 0.55	3.37 ± 1.52	5.47 ± 2.16
**4**	4.64 ± 0.93	3.71 ± 0.40	3.20 ± 0.84	3.36 ± 0.84
**5**	3.54 ± 1.05	3.08 ± 0.26	4.11 ± 1.27	2.13 ± 0.49
**6**	7.54 ± 3.15	10.48 ± 5.61	6.61 ± 2.48	6.04 ± 0.89
**7**	23.33 ± 4.07	- ^2^	- ^2^	- ^2^
**8**	2.50 ± 1.88	4.17 ± 0.19	2.86 ± 0.18	3.35 ± 0.16
**9**	3.72 ± 0.36	4.94 ± 0.69	7.28 ± 2.50	8.58 ± 3.40
**10**	2.24 ± 0.83	2.71 ± 0.12	2.18 ± 1.05	2.89 ± 0.34
cisplatin	2.40 ± 0.32	3.07 ± 0.45	4.01 ± 1.58	3.65 ± 0.76

^1^ IC_50_—compound concentration that inhibits proliferation rate of tumour cells by 50%, compared to untreated control cells. Values are the means ± SD of nine independent experiments. ^2^—studies not carried out.

**Table 4 molecules-27-08511-t004:** Lipophilicity parameters of compounds: log k_w(RP−18)_ and -S values (Equation (1)) obtained by the RP-18 HPLC chromatography (pH = 7.4) and computational methods.

Compound	-S	log k_w_	r^2^	n	Mlog P	log P ^1^	Clog P
**1**	4.358	3.273	0.994	7	1.778	2.26	2.1600
**2**	4.64	3.301	0.998	7	1.522	2.27	2.15
**3**	5.325	2.92	0.999	7	1.337	2.34	1.98
**4**	5.002	3.411	0.988	7	1.586	2.74	2.46
**5**	5.855	3.774	0.996	6	1.829	3.32	2.88
**6**	5.477	4.223	0.987	6	1.586	2.86	2.54
**7**	4.811	2.800	0.990	6	0.577	0.77	1.59
**8**	4.263	3.936	0.988	7	2.641	3.43	3.15
**9**	5.395	4.901	0.996	6	3.111	4.41	4.06
**10**	4.996	4.518	0.994	6	3.146	3.95	3.71

^1^ log P—log *p* values calculated according to Crippen’s fragmentation [50].

## Data Availability

Not applicable.

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
