# Peer review of "Cholinesterases Inhibition, Anticancer and Antioxidant Activity of Novel Benzoxazole and Naphthoxazole Analogs"

_molecules, 2022, doi:10.3390/molecules27238511_

Round 1
Reviewer 1 Report
Review of the Manuscript ID: molecules-2036572 The manuscript entitled „Cholinesterases Inhibition and Anti-cancer Activity of NovelBenzoxazole and Naphthoxazole Analogs” by Alicja Skrzypek et al. reports the synthesis of a series of seven benzoxazoles and three naphtho[1,2-d][1,3]oxazole derivatives, and the studies of their potential biological activities. In the studies, the cytotoxicity of ten synthesized compounds was determined, and for three of them the toxicity against normal keratinocytes and normal mammary gland cells was assessed. In addition, the inhibitory properties of six selected compounds, against acetylcholinesterase and butyrylcholinesterase activities were determined. For two compounds, antifungal activity has been demonstrated. Lipophilicity of all synthesized compounds was examined with chromatographic method. Appreciating the large contribution of the research team, I would like to draw attention to some shortcomings and errors in the description of the research results.
1. First of all, interpretation of the results should be more careful. The conclusion concerning antiproliferative activity of compound 10 is not justified: the difference between IC50 for 10 and IC50 for cisplatin is not statistically significant. The discussion of the results concerning antifungal properties of two selected compounds 1 and 8 is superficial. In the abstract, the authors state: “The compounds exhibited also antifungal activity against drug-resistant clinical isolates of Candida species as well as antioxidant properties”, whereas, not all compounds were examined for antifungal properties, so, this conclusion cannot be generalised. In turn, antioxidant properties of studied compounds are very diverse and approximately 10-20 times weaker than quercetin.
2. Referring to biological assays, the inhibitory activity of studied compounds against cholinesterases seems to be the most important. I would suggest to explain the role of these enzymes in the Alzheimer’s disease in the Introduction and justify why the inhibition of these enzymes is so important.
3. Line 156-157: The sentence is misleading, the intersection of trend lines in the chart field indicates on the inhibition of mixed-type.
4. In the section Materials and Methods, SRB colorimetric assay as the method of cytotoxicity determination could be described in more detail. This assay detects cytotoxicity, death, viability or proliferation of studied cells. The title of section 4.2.1. “SRB cytotoxicity assay”, better reflects what has been measured.
5. It would be better if the abbreviations, such as AChE, BChE, AD, MIC, have been introduced in the first place of appearance.
6. In the caption to the Table 1 is written: **p<0.001, while at the lines 486-487 of the manuscript, there is a sentence concerning ANOVA test and significance level of 5%. The other question is, what values were compared with the use of this statistic test? This question also applies to AChE and BChE columns.
7. In the Figure 2, it would be desirable to mention that hydrogen is present in the positions of 1 and 4 for the compound 2; in the positions 1,3, and 4 for the compound 3, and similarly for the other compounds.
Minor errors:
Some expressions are awkward or grammatically incorrect:
Line 37: ..are major skeletons…
Line 163: DPPH-scavenging activity
Line 190: The structure-activity relationship concerning antiproliferative/cytotoxic activity of studied compounds;
Lines 126-129: The repetition is unnecessary, however it is not clear what differences are compared at p<0.001;
Line 219: protective effect against these cells, change to protective effect on cells;
Line 237: “they” change to “their”
Line 245: important quantity ( important feature?) that influences on the pharmacokinetic properties….
In this review, only a few language mistakes have been listed. For better quality and clarity of the text the professional linguistic help is very advisable.

Reviewer 2 Report
Skrzypek et al. submitted a manuscript, "Cholinesterases Inhibition and Anti-cancer Activity of Novel Benzoxazole and Naphthoxazole Analogs, " about derivatizing the benzoxazole and naphthoxazole and found active against AChE and Candida species.
The strength of the paper is in the development of molecules where authors characterized NMR spectroscopy, followed by AChE and BChE enzyme kinetics.
Comments.
1. The rationality of the paper developing the AChE and BChe looks too generalized; please improve the introduction section.
2. There should be a paragraph outlining the cholinesterase activity in the cancers.
3. Adding antifungal activity after an anticancer study lacks proper justification. Please write a paragraph about why cholinesterase targeting is required with antifungal activity. Does the scientific community require such dual activities? Please explain with rationality.
4. Please properly label the supplementary spectral data, as it is quite confusing to read it.
The paper is written systematically and can be considered if the authors agree to revise the manuscript based on the above comments.
Reviewer 3 Report
Title: “Cholinesterases Inhibition and Anti-cancer Activity of Novel Benzoxazole and Naphthoxazole Analogs”
The authors have written the manuscript in a very sequential and scientific way. This manuscript is well-designed and well-described and covered all necessary parameters. There are many major concerns/flaws and areas which should be improved before publishing it.
1. Title of the study should be based on to cover all the key parameters of the study, in my opinion it should be modified according to the key objective of the study.
2. The abstract should be started with mentioning background of benzoxazole and naphthoxazole analogs whether they have any biological properties previously reported.
3. Abstract should be amended with some numerical values besides from IC50 values from the key results of the study.
4. The last sentence 32, 33 of the abstract should be concluded on a suitable elaboration sentences that attract the readers related to this work.
5. Introduction section; introduction lack description related to cholinesterases, it should be amended and supported by appropriate references as it look did not specify the direction for the readers; for guidance like. https://doi.org/10.3390/molecules27082468, doi: 10.3390/molecules26237168; doi: 10.1007/s00044-019-02497-0;
6. The author should mention at end of introduction the rationale for designing such type of analogs.
7. Results: the author has mixed the discussion with the study results; it should be separate from the results.
8. Resolution of the figure 5 is too low to be readable; its pixel should be enhanced.
9. It would be better to show the results in figure 5 with colors legends, to be read easily.
10. Conclusion of the study is too lengthy, in my opinion it should be based on the key objective of the study and the expressions for future concerns.
11. Synthesis procedure; Line 296 to 302 should be rephrase for better understanding, moreover, this portion should be supported by appropriate references.
12. Line 308; there is typo mistake after IR (KBr, cm-1): “V” this symbol is missing in rest of the compounds, weather it is appropriate or mistakenly added?
13. Conclusions: the author writes the conclusion two time in the manuscript as a mistake after methods sections, it should be deleted from the methods section and retain the previous one.
14. Most of the methods lack updated references.
15. There are various grammatical and typos mistakes found, it should be eliminated.
16. The authors should provide limitation of the study.
17. In my opinion the author should revised the manuscript thoroughly for the mentioned concerns before publishing.
Round 2
Reviewer 1 Report
The authors still should correct the part of the manuscript relating to the enzyme inhibition.
1. First of all, in the Lineweaver-Burk plot, there are other concentrations than these given in the paragraph: “Kinetic studies were performed in the same manner as in the earlier experiments while the substrate (ATCI) was used in the concentration range of 2.5-15 nM [39]. In the plot on X axis, there should be values 1/S.
ATCI is an inhibitor, not a substrate!
2. In the paragraph:
“The examination revealed that Vmax decreased with increasing KM and increasing concentrations of the compound 8, which indicated that this complex inhibits AChE in two distinct ways: competitively forming the EI complex and disrupting the enzyme–substrate inhibitor (ESI) complex in a non-competitive manner [44].”
The phrase “disrupting the enzyme–substrate–inhibitor (ESI) complex in a non-competitive manner” is not clear. In this case, the exact mechanism of non-competitive inhibition could be explained with additional research.
Reference 44 would be adequate if authors want to mention their studies of other compounds as inhibitors of AChE.
“Compound 8, which indicated that this complex inhibits” It sounds as compound 8 was a complex.
Reviewer 2 Report
The authors revised the manuscript accordingly.
Author Response
Thank you very much for thoughtful comments and helpful suggestions.
Reviewer 3 Report
Suggestions/Changes incorporated. Needs final spellings and language check before publication.
Author Response
Answers to Reviewer 3:
- Needs final spellings and language check before publication.
Response: Thank you for your comments. The manuscript was thoroughly checked by the translation specialist to ensure that the text is correctly phrased and free from language and grammatical errors.